# Angiogenic Function of Human Placental Endothelial Cells in Severe Fetal Growth Restriction Is Not Rescued by Individual Extracellular Matrix Proteins

**DOI:** 10.3390/cells12192339

**Published:** 2023-09-23

**Authors:** Lauren Sayres, Amanda R. Flockton, Shuhan Ji, Carla Rey Diaz, Diane L. Gumina, Emily J. Su

**Affiliations:** 1Division of Maternal Fetal Medicine, Department of Obstetrics and Gynecology, University of Colorado School of Medicine, Aurora, Colorado, CO 80045, USA; 2Division of Reproductive Sciences, Department of Obstetrics and Gynecology, University of Colorado School of Medicine, Aurora, Colorado, CO 80045, USA

**Keywords:** fetal growth restriction, umbilical artery Dopplers, human placental endothelial cells, extracellular matrix, angiogenesis

## Abstract

Severe fetal growth restriction (FGR) is characterized by increased placental vascular resistance resulting from aberrant angiogenesis. Interactions between endothelial cells (ECs) and the extracellular matrix (ECM) are critical to the complex process of angiogenesis. We have previously found that placental stromal abnormalities contribute to impaired angiogenesis in severe FGR. The objective of this research is to better characterize the effect of individual ECM proteins on placental angiogenic properties in the setting of severe FGR. ECs were isolated from human placentae, either control or affected by severe FGR, and subjected to a series of experiments to interrogate the role of ECM proteins on adhesion, proliferation, migration, and apoptosis. We found impaired proliferation and migration of growth-restricted ECs. Although individual substrates did not substantially impact migratory capacity, collagens I, III, and IV partially mitigated proliferative defects seen in FGR ECs. Differences in adhesion and apoptosis between control and FGR ECs were not evident. Our findings demonstrate that placental angiogenic defects that characterize severe FGR cannot be explained by a singular ECM protein, but rather, the placental stroma as a whole. Further investigation of the effects of stromal composition, architecture, stiffness, growth factor sequestration, and capacity for remodeling is essential to better understand the role of ECM in impaired angiogenesis in severe FGR.

## 1. Introduction

Fetal growth restriction (FGR) is a pathologic condition whereby a fetus does not meet its growth potential [1]. FGR is a leading cause of perinatal morbidity and mortality and confers risk for lifelong adverse health outcomes [2,3,4]. Outcomes are substantially worse in the severe, early-onset phenotype of FGR, defined by an estimated fetal weight or abdominal circumference less than the third percentile or abnormal umbilical artery Doppler velocities diagnosed before 32 weeks [5,6,7]. When Doppler abnormalities have progressed to absent or reversed umbilical artery velocities, there is high risk for perinatal death [8,9,10,11]. Among survivors, there remains substantial risk for adverse early and long-term outcomes [12,13]. To date, there are no effective prophylactic or therapeutic measures for FGR [1]. Thus, following a diagnosis of severe FGR, clinicians are resigned to increased antenatal surveillance and early delivery in order to minimize exposure to the abnormal intrauterine environment and attendant risk of fetal demise [1]. Advancing our understanding of the pathophysiology of FGR is critical to improving outcomes for pregnant women and their offspring.

Abnormal umbilical artery Doppler velocimetry, an indicator of placental insufficiency, is a key finding in severe FGR [14]. Placentae complicated by FGR specifically with absent or reversed umbilical artery Dopplers demonstrate sparse villous vasculature [15,16,17,18,19,20]. This leads to deficient oxygen and nutrient delivery to the fetus and increased fetal cardiac and metabolic strain, key pathophysiologic causes underlying the risk for adverse outcome [15,17]. This aberrant vasculature results from a number of mechanistic pathways that remain poorly understood [21,22]. In particular, to maintain a healthy vascular bed and function, fetoplacental angiogenesis must increase exponentially in the second half of gestation; in severe FGR, this process has been found to be significantly impaired [16].

Angiogenesis entails the complex process of sprouting a new blood vessel from an existing vessel [23]. Endothelial cells (ECs) that line the original vessel must appropriately interact with the surrounding extracellular matrix (ECM), resulting in proliferation, outward migration, and luminal formation. The role of the ECM as a critical source of angiogenic regulatory factors and as a mediator of EC function in other human tissues has been established [24,25]. Ongoing research has suggested that specific characteristics of the ECM, such as protein composition and stiffness, may similarly impact placental angiogenic function [26,27,28,29].

Our laboratory previously demonstrated that human ECs primarily isolated from placentae affected by severe FGR exhibited impaired angiogenesis relative to ECs derived from control placentae [30]. Ongoing efforts have focused on the interaction between ECs and their microenvironment [29]. This work reveals that when ECs obtained from control placentae are plated on cell-derived matrix (CDM) generated from fibroblasts of growth-restricted placentae, these control ECs exhibit impaired angiogenic properties. In contrast, the inherent angiogenic defects of FGR ECs can be partially rescued when plated on control CDM. Together, these underscore the importance of matrix composition on regulation of EC function. Further interrogation of FGR CDM has also suggested a deficiency in expression and deposition of certain stromal proteins, including fibronectin and collagen I. 

The objective of this study is to characterize the effects of individual ECM proteins on the angiogenic properties of human placental ECs in severe FGR with abnormal Doppler velocimetry. Specifically, we have evaluated adhesion, proliferation, migration, and apoptosis of ECs derived from FGR and control placentae in the presence of several ECM proteins [29,31]. The individual proteins—fibronectin; collagens I, III, and IV; laminin; thrombospondin; and fibrinogen—were chosen as prior efforts have revealed these to be abundant in the placental stromal matrix and because they are plausible mediators of fetoplacental angiogenic function [29,31,32]. Our central hypothesis is that the impaired angiogenesis characteristic of severe FGR will be partially rescued in the presence of certain ECM proteins including collagen I.

## 2. Materials and Methods

### 2.1. Subject Selection

This study was approved by the Colorado Multiple Institutional Review Board (COMIRB study IDs 14-1073 and 15-0365). Eligible subjects were identified and approached by our institution’s Perinatal Research Core. All subjects provided informed consent to participate. 

All subjects were patients of the obstetric service at a tertiary academic hospital. Subjects affected by FGR had an estimated fetal weight or abdominal circumference below the 10^th^ percentile with absence or reversal of umbilical artery diastolic flow diagnosed before 32 weeks. Fetal growth was estimated using the Hadlock algorithm. Fenton birthweight percentiles were used to confirm normal or restricted growth. Control subjects delivered a normally grown neonate at term. All deliveries occurred via unlabored cesarean section. Exclusion criteria included use of in vitro fertilization for conception, multiple gestation, fetal anomaly or aneuploidy, maternal infection, or maternal condition that could result in uteroplacental insufficiency, such as diabetes, antiphospholipid antibody syndrome, or substance use. Of note, any form of hypertension, including preeclampsia, was not considered an exclusion criterion for the FGR cohort, as the vast majority of subjects with severe FGR at our institution exhibited a concomitant diagnosis. 

### 2.2. EC Isolation and Culture

Placentae were obtained immediately following delivery. Primary fetoplacental villous macrovascular ECs were isolated using previously described methods [29]. ECs were cultured in microvascular endothelial cell growth medium containing 5% fetal bovine serum, bovine brain extract, human epidermal growth factor, hydrocortisone, gentamicin, and amphotericin B (Lonza Bioscience; Walkersville, MD, USA), except where noted. ECs were cultured in a humidified incubator at 37 °C with 5% CO_2_. ECs between the third and fifth passages were used in all experiments, and passages were matched as best as possible within each experiment in order to mitigate potential effects of passage on phenotype. Technical triplicates were performed except where noted. Figure 1 depicts the experimental design.

### 2.3. EC Adhesion

CytoSelect Cell Adhesion Assay Kit (Cell Biolabs; San Diego, CA, USA) was used in accordance with the manufacturer’s instructions to evaluate EC adhesion to various ECM substrates, including fibronectin, collagens I and IV, laminin, and fibrinogen. ECs from control and severe FGR placentae were suspended at a concentration of 5 × 10^5^ cells/mL in serum-free endothelial cell growth medium (Lonza Bioscience). The cell suspension was applied to each substrate-coated well and incubated at 37 °C for 60 min. Each well was serially washed, and cell stain and extraction solutions were sequentially applied. Samples were transferred to a microtiter plate, and optical density at 560 nm was determined using a plate reader. Colorimetric measurements were standardized to those of both negative experimental controls (wells coated with bovine serum albumin) and negative technical controls (no cells added to the wells).

### 2.4. EC Proliferation

Live cell microscopy was used to determine EC proliferation on various ECM substrates. Fibronectin; collagens I, III, and IV; laminin; and thrombospondin were diluted in phosphate-buffered solution (PBS) to a concentration of 10 μg/mL. Equal volumes of 30 μL of each substrate were applied to individual wells of a microtiter plate and allowed to coat the wells over two hours on an orbital shaker. The wells were then washed, and ECs were plated at a density of 2.5 × 10^3^ cells/mL into each well. The plates were incubated at 37 °C for 5 d, with live cell microscopy performed every 24 h, including at time of initial plating. ECs were counted using calibrated software, Incucyte Live Cell Analysis (Sartorius, Öttinger, Germany). A total of four technical replicates were performed per subject for each ECM substrate. This particular experiment was performed on cells from 14 subjects, 7 controls and 7 with severe FGR.

### 2.5. EC Migration

Migration of ECs in the presence of various ECM substrates was determined via wound scratch assay. Similar to the proliferation experiment, equal volumes and concentrations of fibronectin; collagens I, III, and IV; laminin; and thrombospondin were individually coated on a microtiter plate. After a two-hour incubation, the wells were washed, and ECs were plated at a density of 5.5 × 10^4^ cells/mL into each well. The ECs were incubated at 37 °C for 18 h, ensuring 100% confluence was achieved. A wound scratch was then performed with the Incucyte WoundMaker (Sartorius), and the wells were washed with culture medium to remove debris. Live cell microscopy was performed at baseline, confirming initial confluence of the cells and quality of the wound scratch, and every three hours for a total 24 h period. Following iterative calibration, Incucyte Live Cell Analysis was performed to determine the relative wound density. Relative wound density is defined as the proportion of the initial wound scratch occupied by cells at a given time point and accounts for temporal changes in the background, non-wound, and wound density. A total of at least three technical replicates were performed per subject for each ECM substrate.

### 2.6. EC Apoptosis

Apoptosis of ECs in the presence of each ECM substrate was determined by performing automated capillary immunoblotting for cleaved caspase-3, a protein in the apoptosis pathway, and total caspase-3 as a positive control. Individual plates were coated with 1 mL of 10 μg/mL fibronectin; collagens I, III, and IV; laminin; and thrombospondin and allowed to incubate for two hours. Plates were washed, and 5 × 10^4^ cells/mL were applied to each plate. After a 5 d incubation at 37 °C, the ECs were washed with PBS and detached from the plates using trypsin. The ECs were serially centrifuged with culture medium then PBS followed by aspiration of the supernatant after each step. Next, 70 μL of Mammalian Protein Extraction Reagent (M-PER™) containing protease and phosphatase inhibitors (Thermo Fisher Scientific; Waltham, MA, USA) was applied to each tube of ECs. Protein extraction was allowed to occur over a 20 min period by maintaining the cell solution on ice with frequent vortexing. The extracted proteins were then centrifuged and stored at −80 °C for later use. As a positive apoptotic control, ECs that had been treated with 5% ethanol for 30 min underwent the same protein extraction process.

Protein concentration was calculated using a standard bicinchoninic acid protein assay kit (Thermo Fisher Scientific). Automated Jess capillary-based immunoblotting (Bio-Techne; Minneapolis, MN, USA) was performed to detect cleaved and total caspase-3 in accordance with the manufacturer’s instructions. Protein samples were loaded into the microplate at a concentration of 0.3 μg/μL. Primary rabbit antibodies to cleaved caspase-3 at a dilution of 1:10 (Cell Signaling Technologies; Danvers, MA, USA, 9661s) and total caspase at a dilution of 1:500 (Cell Signaling Technologies; 9662s) were selected based on our prior optimization efforts [29]. Expected molecular weights are approximately 17 and 19 kDa for cleavage products of caspase-3 and 36 kDa for total caspase-3 [33]. A total protein normalization package (Bio-Techne) was employed to ensure consistent protein loading across samples. Single replicates were run for each EC phenotype in the presence or absence of each ECM substrate given consistent findings across all samples and subjects. Both negative and positive apoptosis controls were included in each Western blot run to ensure accurate interpretation.

### 2.7. Statistical Analysis

An a priori power calculation was performed, demonstrating that eight subjects per group were required for 90% power (α = 0.05) to detect a biologically significant 10% difference in average migratory capacity. Shapiro–Wilk testing did not suggest non-Gaussian distributions for outcomes with continuous variables for either control or FGR ECs. Clinical characteristics between control and FGR subjects were compared using Fisher’s exact test, student’s *t* test, or Welch’s *t* test, as applicable. EC adhesion, proliferation, and migration were compared using two-way analysis of variance (ANOVAs) followed by Šidák’s tests when applicable. EC adhesion based on substrates was also compared using a two-way ANOVA followed by Tukey’s multiple comparisons tests when applicable. EC apoptosis was assessed by evaluating the area under the curve at the expected molecular weights for cleaved and total caspase-3 on Western blot images. Alpha significance level was set to a p value of less than 0.05. Error bars in all figures represent the standard error of the mean. Commercially available statistical software (GraphPad Prism 9; GraphPad Software; Boston, MA, USA) was used.

## 3. Results

### 3.1. Subject Characteristics

Demographic and clinical characteristics of all subjects are presented in Table 1. Included are a total of 16 subjects, of whom 8 were diagnosed with severe FGR and 8 represented term controls. As expected, there were significant differences between control and severe FGR subjects with regard to gestational age at delivery (39 v. 28 weeks, *p* < 0.0001), neonatal birth weight (3220 v. 647 grams, *p* < 0.0001), and Fenton percentile (43 v. 3, *p* = 0.002). Nulliparity was less common among control subjects relative to those diagnosed with FGR (0% v. 38%, *p* = 0.03). While there was no significant difference in proportion of fetal sex between the two cohorts, there was a slight skew toward more male fetuses with FGR (five male, three female) as compared to control subjects, which were evenly distributed (four male, four female). All subjects with FGR displayed absent or reversed umbilical artery Doppler flow; no control subjects had this finding. 

### 3.2. EC Adhesion

First, we sought to determine whether adhesion of ECs varies in the setting of different ECM substrates. We evaluated adhesion of control and severe FGR ECs to five ECM proteins: fibronectin, collagen I, collagen IV, laminin, and fibrinogen. We found no difference in the adhesion of control relative to FGR ECs on each individual substrate (Figure 2). There were also no sex-specific differences in adhesive capacity (Appendix A). However, when comparing adhesion of all ECs by ECM protein (Figure 2), significant differences were found. Relative to laminin, adhesion of all ECs to fibronectin (*p* = 0.001), collagen I (*p* = 0.0002), and collagen IV (*p* < 0.0001) was significantly increased. Similarly, as compared to fibrinogen, ECs from both cohorts also displayed enhanced adhesion to collagen I (*p* = 0.02) and collagen IV (*p* = 0.009) (Figure 2). Together, these data suggest that although there is no difference in adhesion between control and FGR ECs, ECs still exhibit preferential binding to certain stromal ECM proteins.

### 3.3. EC Proliferation

Next, we compared the growth of control and FGR ECs when plated on each of six ECM substrates—fibronectin, collagens I, III, and IV, laminin, and thrombospondin—and on uncoated plates. Overall, when combining all substrate and uncoated conditions, there was significantly less proliferation in FGR ECs as compared to control ECs (*p* = 0.002), with post hoc comparisons demonstrating significant differences at days 3 (*p* = 0.003), 4 (*p* = 0.0007), and 5 (*p* = 0.01) (Figure 3A). However, there were no significant differences between control and severe FGR EC proliferation once they were stratified by substrate (Figure 3B). Further analysis demonstrated that as compared to lack of ECM substrate, collagen I, collagen III, and collagen IV promoted FGR EC proliferation (Figure 3C). In contrast, there was no statistical improvement in control EC proliferative capacity in the setting of any ECM protein (Figure 3C). This suggests the possibility of a differential response of control and FGR ECs to certain ECM proteins, such that these proteins may partially rescue proliferative capacity in severe FGR. When analyzed by fetal sex, proliferation curves were not significantly different between male and female subjects (Appendix A).

### 3.4. EC Migration

Our next step was to evaluate the migration of third or fourth passage ECs across a wound scratch over a 24 h period in the presence or absence of ECM proteins. With all substrates combined, FGR ECs exhibited decreased migration as compared to control ECs (*p* < 0.0001) (Figure 4A; representative images of substrates with statistically significant differences shown in Appendix A). Post hoc comparisons demonstrate significant differences at 3, 6, 9, 12, 15, and 18 h (*p* < 0.0001 for all timepoints) and 21 h (*p* = 0.01). FGR ECs continued to demonstrate significantly decreased migratory capacity in the absence of ECM substrate (*p* = 0.04) or in the presence of fibronectin (*p* = 0.04), collagen I (*p* = 0.048), or laminin (*p* = 0.03) (Figure 4B), suggesting that these substrates do not substantially alter control or FGR EC motility. Furthermore, there were no significant differences at each individual timepoint for any of these substrates with post hoc testing. With regard to collagen III, collagen IV, and thrombospondin, there were no significant differences between control and FGR EC migratory capacity. However, further comparison of either control or severe FGR ECs in the presence of each substrate relative to migration on uncoated plates did not demonstrate any significant differences in either cohort (Figure 4C). Together, our data indicate that no specific ECM protein rescues FGR EC migration. Notably, both control and severe FGR ECs isolated from placentae of female fetuses exhibited significantly improved migration relative to males (Appendix A). In fifth passage ECs, control and FGR ECs on uncoated plates displayed similar rates of migration (*p* = 0.37, Appendix A). This finding persisted in the presence of all individual substrates (Appendix A), suggesting that passage number may impact specific angiogenic properties.

### 3.5. EC Apoptosis

Finally, we sought to assess whether apoptosis of ECs differed in the presence of individual ECM substrates. We performed automated capillary immunoblotting on proteins extracted from control and severe FGR ECs cultured for five days on substrate-coated or uncoated plates. We detected the expression of total caspase-3, an expected protein in live cells, across all control and FGR ECs cultured in the presence or absence of any ECM substrates. However, cleaved caspase-3, a marker of apoptosis, was not detected in either control or FGR ECs when cultured with or without individual ECM substrates as compared to a positive apoptotic control of fetoplacental ECs treated with 5% ethanol for 30 min (Figure 5). All Western blots produced identical results with complete absence of cleaved caspase-3 in all control or FGR EC subjects regardless of substrate, with the exception of the positive control. Summarily, these results demonstrate that apoptosis was not induced in either cohort regardless of ECM substrate and, thus, cannot explain our other findings of differential adhesion, proliferation, or migration.

## 4. Discussion

Placental vascular development is critical to its function in delivery of oxygen and nutrients to the fetus and fetal cardiac dynamics. In particular, inappropriately sparse vasculature is evident in severe FGR and associated with morbidity and mortality. Placental ECs do not operate in vacuo but rather are surrounded by a complex extracellular environment that affects their angiogenic potential [29,34]. We previously demonstrated the existence of a unique ECM signature among FGR placentae, with decreased, disordered deposition of certain stromal proteins, including fibronectin and collagen I, but no difference in thrombospondin-1, in FGR CDM [29]. However, subsequent work suggested no significant improvement in severe FGR EC migratory capacity in the presence of fibronectin [35]. Thus, we hypothesized that exposure of severe FGR ECs to other ECM proteins, specifically collagen I, would partially rescue the impaired angiogenic properties seen at baseline. 

Our overall findings are consistent with prior work demonstrating that there is an intrinsic defect in the proliferation and migration of ECs derived from placentae affected by severe FGR [29]. We further demonstrate that collagens I, III, and IV partially rescue severe FGR EC proliferation. However, no individual substrates appeared to mitigate FGR EC migratory deficiencies that were evident in earlier passage cells. In contrast, our findings of similar migration between control and severe FGR ECs at a later ex vivo passage run counter to prior research. While our laboratory had previously validated other fetoplacental EC properties through the fifth passage, including angiogenic tube formation, this had not previously been completed for wound scratch assays. Other groups have also shown stability of placental and other types of ECs at the cellular, molecular, and transcriptional levels through the fifth passage [36,37,38]. However, we speculate that passaging our cells may specifically alter the dynamics and function of EC surface receptors such as integrins, which bind to ECM proteins and play a critical role in cellular migration. Although not specific to ECs, it has been demonstrated that a higher passage number of various cell types results in differential integrin subcellular localization and rates of trafficking [35,39,40,41].

We did not uncover evidence of differential adhesion among placental ECs, either control or FGR. It is apparent that adhesion and apoptosis of other placental cells plays a critical role in obstetric pathology including FGR [35,42,43,44]. The fact that ECs adhered differently to various ECM substrates points to the critical role of adhesion in angiogenic function and suggests that there might be other ways in which adhesion differs, particularly given the unique ECM patterning in severe FGR. With regard to apoptosis, we did not find any evidence of cleaved caspase-3 in control or FGR ECs regardless of substrate. This suggests that excessive apoptosis is not a primary mechanism underlying impaired angiogenesis in severe FGR. However, it is certainly plausible that the complexities of ECM (that are not recapitulated by individual substrates) may induce either intrinsic or extrinsic apoptotic signaling pathways.

While we were not powered to detect sex-specific differences, we did see evidence of improved EC migration (both control and severe FGR) in cells isolated from placentae of female compared to male fetuses. Interestingly, these sex-specific differences were not evident in adhesion or proliferation. These novel findings suggest that fetal sex may mediate certain angiogenic properties of fetoplacental ECs in severe FGR. Future investigation with sample sizes powered to detect sex-specific differences is warranted.

It is well-established that ECM biochemically and structurally impacts cellular organization and physiological functions. For instance, on an individual protein level, integrin–collagen I interactions in dermal microvascular ECs have been shown to promote cell survival and inhibit apoptosis [45]. Furthermore, ligand binding of collagen I-specific integrins results in down-regulation of signaling pathways, induction of actin polymerization, and formation of prominent stress fibers, which are required for cellular migration [46]. Notably, however, these findings were specific to ECs and were not seen in fibroblasts exposed to collagen I [46]. This suggests that matrix effects are specific to cell type and further support the importance of ECM, in particular, on angiogenesis.

A strength of our research is the use of clinically relevant, primarily isolated human placental cells from well-validated subjects with clear FGR or control phenotypes. However, this study is limited by its focus on the effects of individual exogenous ECM substrates and the use of a single but uniform concentration of each individual ECM protein. Future directions should include evaluation of EC angiogenic properties in the setting of different combinations and patterning of ECM proteins. The healthy placental ECM has been previously characterized through data-mining of a cDNA library, and a more extensive, proteomic cataloging of the ECM composition in uncomplicated and severe FGR placentae may also better guide future investigations into candidate proteins [47]. Additionally, other characteristics of the placental ECM, such as mechanical stiffness, which is increased in severe FGR, should be assessed for their angiogenic consequences [48,49]. As EC–ECM interactions are bi-directional, investigational efforts should also be undertaken to characterize how FGR ECs in turn regulate ECM composition [23]. Finally, angiogenesis is a dynamic process across placental development, and exploration of the effects of the ECM on EC properties at different gestational ages should be undertaken [26].

In summary, we have detected diminished proliferation and migration of fetoplacental ECs in severe FGR. This impaired angiogenic profile is not predicated on a single ECM protein, although collagens I, III, and IV may actually promote severe FGR EC migration. This confirms that the complexities of placental villous stroma, ranging from composition, architecture, stiffness, growth factor sequestration, capacity for remodeling, and sex-specific characteristics, all require further investigation. A detailed comprehension of the impaired angiogenesis characteristic of severe FGR is essential to identifying preventative and treatment strategies of this important obstetric pathology.

## Figures and Tables

**Figure 1 cells-12-02339-f001:**
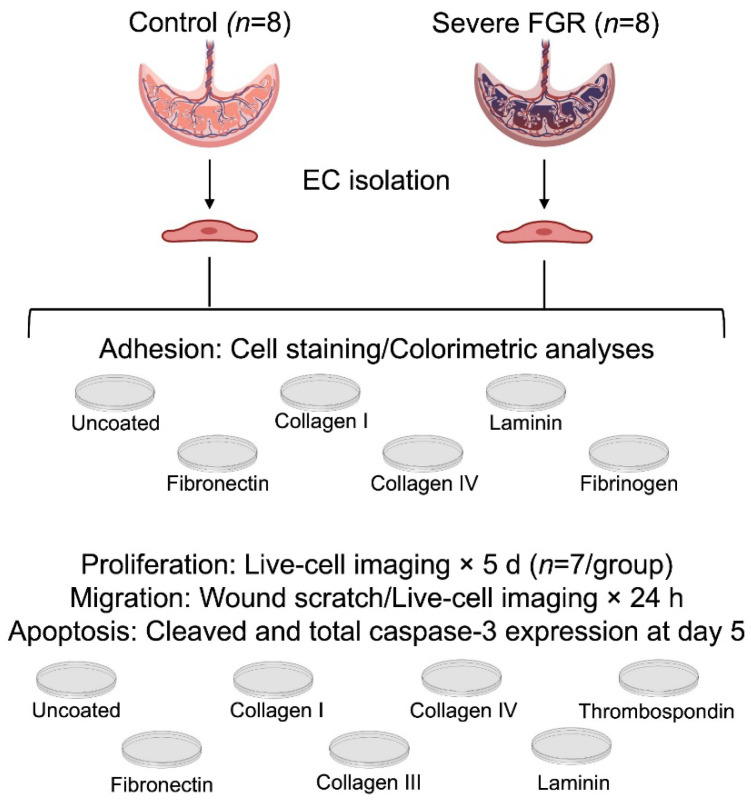
Experimental design. All experiments were performed with a minimum of technical triplicates. There were 8 subjects in each group unless otherwise indicated.

**Figure 2 cells-12-02339-f002:**
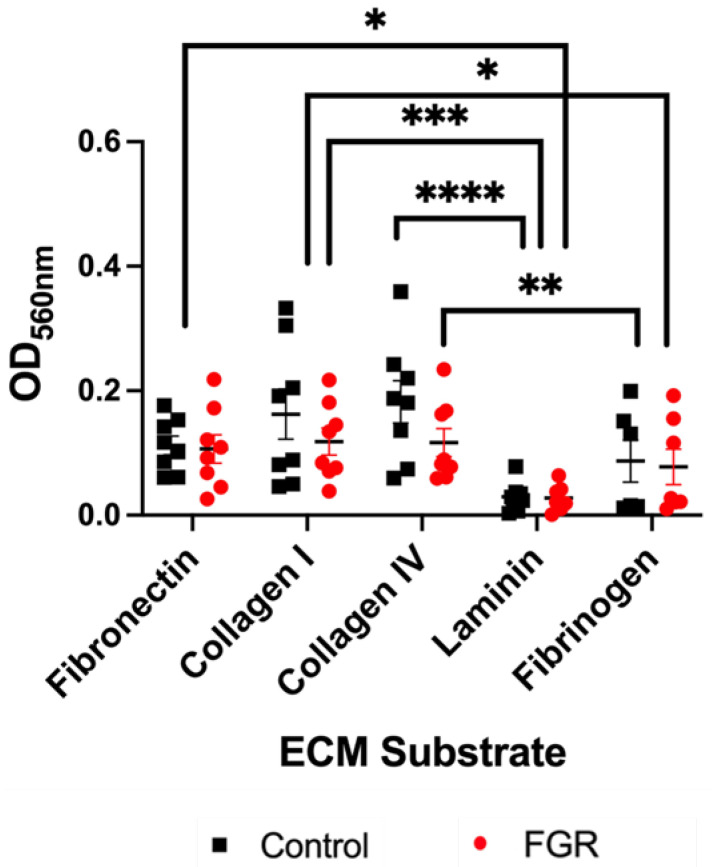
EC adhesion to ECM substrates is not different between control and severe FGR phenotypes. Graphical representation of EC adhesive properties to ECM substrates shows that adhesion to any substrate is not significantly different between control (*n* = 8) and severe FGR (*n* = 8) ECs. Statistical differences detected between substrates are indicated (* *p* < 0.05, ** *p* < 0.01, *** *p* < 0.001, **** *p* < 0.0001). OD: optical density.

**Figure 3 cells-12-02339-f003:**
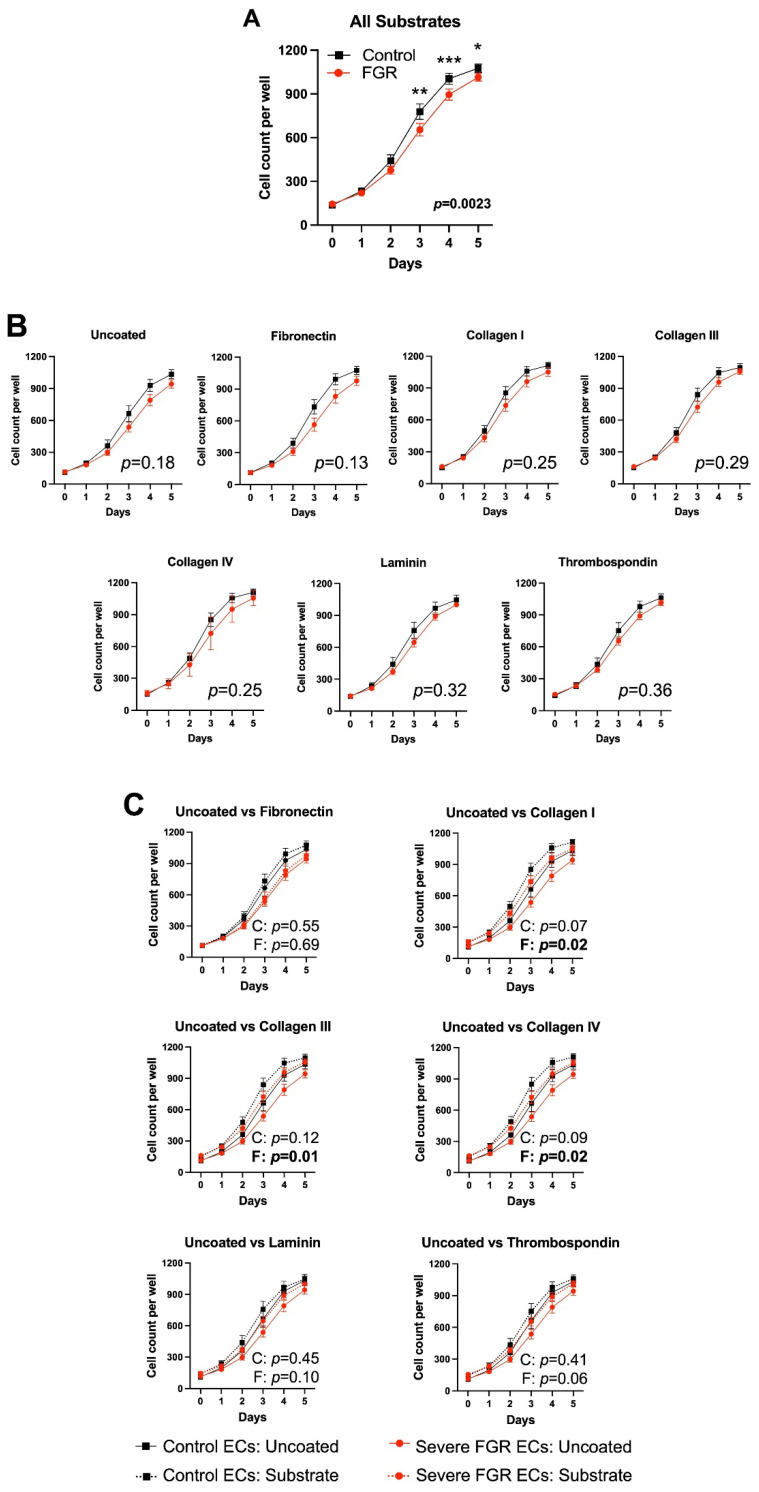
Control ECs proliferate more robustly than their growth-restricted counterparts, although collagens I, III, and IV appear to partially mitigate severe FGR EC proliferation. (**A**) Overall, control ECs (*n* = 7) demonstrate increased proliferation over five days relative to severe FGR cells (*n* = 7), with statistical differences at days 3, 4, and 5 on post hoc comparison (* *p* < 0.05, ** *p* < 0.01, *** *p* < 0.001). (**B**) This difference in proliferation was statistically negated once stratified by individual substrate. (**C**) Comparison of individual substrates to uncoated conditions in both control and severe FGR ECs show that only collagens I, III, and IV partially mitigate FGR EC proliferative impairment (C: *p* values comparing each individual ECM substrate to lack of substrate for control ECs; F: *p* values comparing each individual ECM substrate to lack of ECM protein for severe FGR ECs). Statistically significant findings are bolded. Data represent four technical replicates per each of the 14 subjects.

**Figure 4 cells-12-02339-f004:**
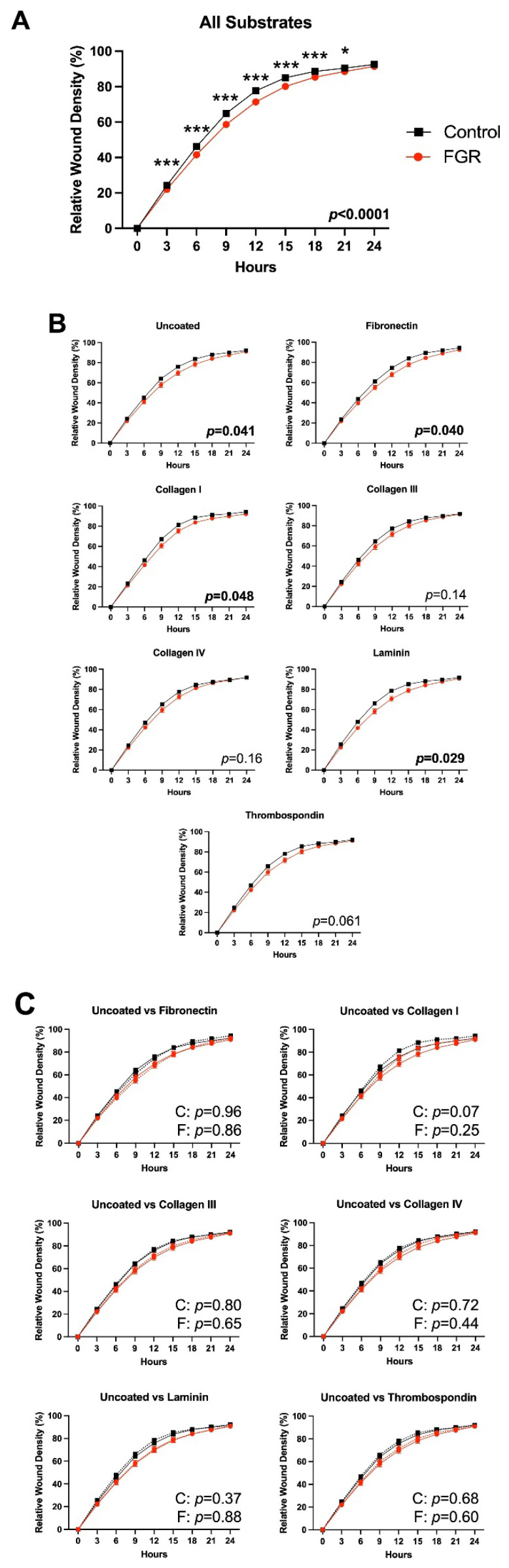
Growth-restricted ECs migrate more slowly than control ECs, and this effect is not rescued by any individual ECM substrate. (**A**) Control ECs (*n* = 8) at passages 3–4 exhibit improved migratory potential relative to severe FGR cells (*n* = 8), with post hoc statistical differences at all time points except for 0 and 24 h (* *p* < 0.05, *** *p* < 0.001). (**B**) This impairment in FGR EC migration is persistent in the setting of no ECM protein, fibronectin, collagen I, and laminin. (**C**) Migration of control or severe FGR EC in the setting of individual ECM substrates was compared to migration in the absence of substrate. Fibronectin, collagen I, and laminin do not enhance FGR EC migration. Neither control nor FGR EC migration was significantly altered in the presence of any individual ECM substrate. (C: *p* values comparing each individual ECM substrate to lack of ECM protein for control ECs; F: *p* values comparing each individual ECM substrate to lack of ECM protein for severe FGR ECs). Statistically significant findings are bolded. Data represent four technical replicates per each of the 16 subjects.

**Figure 5 cells-12-02339-f005:**
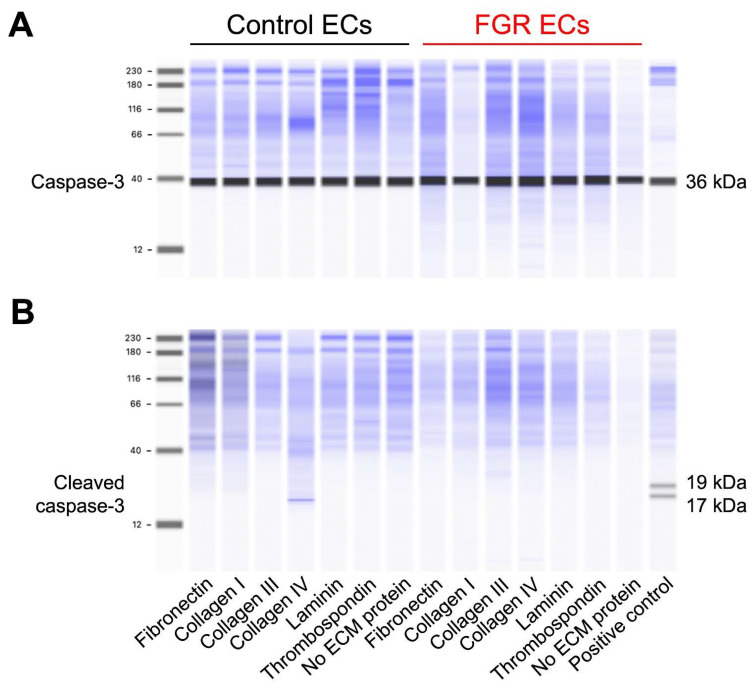
There is no evidence of apoptosis for control or severe FGR ECs cultured in the presence or absence of any individual ECM substrate. (**A**) Capillary immunoblot demonstrates that total caspase-3, represented by the 36 kDa marker, is expressed in all ECs (*n* = 8 for control and *n* = 8 for severe FGR) regardless of the presence or absence of individual ECM substrates. (**B**) Cleaved caspase-3, a marker of apoptosis and detected at 17 and 19 kDa as denoted by the positive control, was not detected among all ECs and all substrates. Total protein normalization within the capillary immunoblots is shown in blue. All immunoblots for each of the 16 subjects produced identical results with presence of total caspase-3 and absence of cleaved caspase-3, with the exception of the positive control for apoptosis.

**Table 1 cells-12-02339-t001:** Demographic and clinical characteristics.

	Control (*n* = 8)	Severe Fetal Growth Restriction (*n* = 8)	Difference (*p* Value)
Maternal age (years)	34 (26–39)	32 (23–38)	0.34
Nulliparity	0 (0%)	3 (38%)	**0.03**
Gestational age at delivery (weeks^days^)	39^1^ (39^0^–39^3^)	28^3^ (24^3^–34^1^)	**<0.0001**
Neonatal sex (female)	4 (50%)	3 (38%)	0.85
Birth weight (grams)	3220 (2898-3840)	647 (370–1425)	**<0.0001**
Fenton percentile	43 (14–85)	3 (1–7)	**<0.01**

Presented as mean (range) or *n* (%). Statistically significant differences indicated with bold font.

## Data Availability

Data supporting the reported results will be made available upon request to the corresponding author.

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
