# Peer review of "Angiogenic Function of Human Placental Endothelial Cells in Severe Fetal Growth Restriction Is Not Rescued by Individual Extracellular Matrix Proteins"

_cells, 2023, doi:10.3390/cells12192339_

Round 1

Reviewer 1 Report

This study aims to determine the roles of several extracellular matrix proteins in human placental endothelial dysfunction in fetal growth restriction.

Major Concerns:

1.     Please include the IRB approved protocol # in 2.1. Subject Selection.

2.     Is there any difference in female and male placental endothelial cell function and responses to the different extracellular matrix proteins?

3.     Fig.1 It would be easier to follow if the author marked out the significant differences in Fig.1A and removed all non-significant P values in Fig.1A. Fig.1B is a table and is not necessary if the comparisons are marked out in Fig.1A.

4.     Please clarify the P-values in figure 2 and 3 are for which time point? What statistical test(s) was/were used?

5.     Data presented in Fig.2A and Fig.3A are difficult to compare with the data presented in Fig.2B and 3B. The current layout makes it very difficult to tell how much recovery each treatment is having and if the recovery effect is statistically significant. It is necessary to plot all comparison groups in the same panel to show the differences.

Minor Concerns:

1.     Line 36-37 “Clinical evidence is clear … there is high risk for perinatal death.” Reference missing

2.     Line 46-47 Placentae from pregnancies complicated …villous vasculature.” Reference missing

Author Response

Thank you for your review. Your comments have helped us substantially improve this manuscript. Responses to your individual comments are below.

Major concerns

  1. Please include the IRB approved protocol # in 2.1. Subject selection.
  • We apologize for the omission. The two relevant IRB protocol numbers have been included.

  1. Is there any difference in female and male placental endothelial cell function and responses to the different extracellular matrix proteins?
  • Thank you for bringing up this very important point. Our sample size was initially chosen based on an a priori calculation for 90% power (a=0.05) to detect a biologically significant 10% difference in average migratory capacity, which showed that n=8 per group was required. While we were able to evenly balance neonatal sex with 4 males and 4 females in the control group, there was a slight skewing, with 5 males and 3 females in the FGR cohort. As we isolate every severe FGR placenta that is eligible, we are unfortunately unable to specifically recruit FGR subjects for even sex-matching.
  • Our initial rationale not to perform additional sex-specific analyses was based on concerns of inadequate power with these sub-group queries. Furthermore, we had prior preliminary data suggesting there were no overt sex-based differences. However, your insightful question led us to reassess this. We analyzed adhesion, proliferation, and migration in a sex-dependent manner. There were no sex-specific differences in adhesion or proliferation. In contrast, both control and FGR ECs from placentas of female neonates exhibited significantly improved migration as compared to control and FGR ECs from male placentas, respectively. We have substantially modified the results, and the sex-specific differences are submitted as supplemental figures. We also call for future lines of inquiry into this question of sex-specific differences in placental angiogenesis.

  1. Figure 1 would be easier to follow if the author marked out the significant differences in Figure 1A and removed all non-significant p-values in Figure 1A. Figure 1B is a table and is not necessary if the comparisons are marked out in Figure 1A.
  • We agree with your comment. We had initially attempted to show all the significant differences, which were limited only to substrate differences, in Figure 1A (bar graph). To more definitively show that there is no significant difference between control and FGR EC adhesion to each of the individual substrates, we have altered our graph from a bar graph to one that shows individual data points, which we believe makes it easily evident that significant differences are limited to substrate. We have also added these comparisons and updated our figure legend to more clearly reflect this.

  1. In figures 2 and 3, which time point do the p-values signify? What statistical test(s) was/were used?
  • We apologize for our original lack of clarity. The p-values listed were for non-linear regression curves and not individual timepoints. We had felt that the overall rates of proliferation and wound closure were more biologically relevant than the potential for significant differences at specific time points.
  • That being said, based upon your and other reviewer critiques, we have re-analyzed proliferation and migration data using two-way ANOVA with repeated measures. This has led to some specific differences in the results as compared to our original submission, and we have revised the methods, results, associated figures, and discussion accordingly.

  1. Data presented in Figure 2A and 3A are difficult to compare with the data presented in Figures 2B and 3B. The current layout makes it very difficult to tell how much recovery each treatment is having and if the recovery effect is statistically significant. It is necessary to plot all comparison groups in the same panel to show the differences.
  • Thank you for this helpful comment. Based on re-analyses with two-way ANOVA with repeated measures and revised figures, we hope our data are more easily interpreted.

Minor concerns

Line 36-37. Reference is missing for “Clinical evidence is clear…there is high risk for perinatal death.”

Line 46-47 Reference is missing for “Placentae from pregnancies complicated…villous vasculature.”

  • We apologize for this oversight. We have amended the bibliography to reflect these changes.

Reviewer 2 Report

In this article, the authors investigated the role of several ECM proteins, including fibronectin, collagen I, III, and IV, laminin, thrombospondin, and fibrinogen in human placental endothelial cells in fetal growth restriction compared to healthy control. Please see below my specific comments and concerns:

1.      My main concern is that much of the conclusions that the authors draw from the presented are from visual interpretations or very brief quantitative analyses. More quantitative analyses and clarification on the statistical methods used to support the conclusions are needed to substantiate the conclusions.

2.      Can the authors please explain the p-values reported in Figs. 2-3? Are they all for a specific time point (5 day)? Are there significance in earlier timepoints? What test was used?

3.      The conclusion on apoptosis seems to be drawn just from visual assessments of western blots. Is there a loading control? Can the authors include some of the full blots in the supplement? Could they perform some quantitative analysis and statistical analysis to support the conclusion?

4.      Its not clear to me how the results presented in Fig 3C supports the conclusion that the response to collagen I is blunted in the FGR group.

Author Response

Thank you for your review. Your comments have helped us substantially improve this manuscript. Responses to your individual comments are below.

  1. My main concern is that much of the conclusions that the authors draw are from visual interpretations or very brief quantitative analyses. More quantitative analyses and clarification on the statistical methods used to support the conclusions are needed to substantiate the conclusions.
  • Thank you for this important comment. These issues have been addressed, with changes reflected in methods, results, discussion, and the figure. Some additional details are also provided below in the responses to this reviewer’s additional concerns.

  1. Can the authors please explain the p-values reported in Figures 2 and 3? Are they all for a specific time point (5 day)? Are there significant differences in earlier time points? What test was used?
  • We apologize for our original lack of clarity. The p-values listed compared the proliferation and wound scratch curves based on non-linear regression curves and did not assess individual time points. We had believed that the overall rates of proliferation and wound closure were more biologically relevant than the potential for significant differences at specific time points.
  • However, we appreciate your comment, and based on this as well as other reviewer critiques, we have re-analyzed proliferation and migration data using two-way ANOVA with repeated measures. This has led to some specific differences in the results as compared to our original submission. The methods, results, associated figures, and discussion have been revised, accordingly.

  1. The conclusion on apoptosis seems to be drawn just form visual assessments of western blots. Is there a loading control? Can the authors include some of the full blots in the supplement? Could they perform some quantitative analysis and statistical analysis to support the conclusion?
  • We are sorry that we did not clearly describe apoptosis findings in our initial manuscript. Our goal had been to depict that there was absence of cleaved caspase-3 in all of the conditions (except the positive control), demonstrating the absence of apoptosis under all substrate conditions. In contrast, total caspase-3 was present, as anticipated, in all specimens. As total caspase-3 was expressed in all lanes, we did not initially show the total protein normalization that is part of the JESS SimpleWestern capillary immunoblot system. However, we have added this to both the cleaved and total caspase-3 images.

  1. It is not clear to me how the results presented in Figure 3C support the conclusion that the response to collagen I is blunted in the FGR group.
  • Thank you for this comment. This was not specifically shown in Figure 3C. In response to reviewer critiques and statistical re-analysis, we have modified our results section and figures.

Reviewer 3 Report

This article is about analyzing the effect of extracellular factors such as laminin, collagen and thrombospondin in primary placental endothelial cells in relation to fetal growth restriction. Extracellular factors are already known from studies such as proteomic studies and the relationship between these proteins and fetal growth restriction has been previously shown. However, this study is the first to show us the effect of these extracellular proteins on primary placental endothelial cells. The authors found that only collagen proteins were associated with this disorder. 

Author Response

Thank you for your review. There were no specific questions or comments requiring a response.

Reviewer 4 Report

The submitted manuscript aims to characterize the effects of individual Extracellular Matrix proteins on the angiogenic properties of human fetoplacental Endothelial cells (ECs) in severe fetal growth restriction (FGR). The topic is vital since FGR is a leading cause of perinatal morbidity and mortality.

Overall The manuscript is well-written and easy to read. Although the experiments are well-designed, a few limitations should be addressed to improve the quality of the work.

Below I will discuss my main comments and suggestions, which hopefully can help the authors improve the study.

- The migrational and angiogenic properties of ECs should be investigated extensively to better elucidate how extracellular matrix proteins regulate angiogenesis beyond just the proliferation activity. For example, a relative angiogenic gene expression, and morphological characterization of vasculogenic networks, should all be considered to strengthen this work.

- The same group previously reported that integrin dysregulation contributes to the reduced angiogenic potential of FGR cells. Why were integrins not evaluated in the present study?

- For the wound scratch assay, representative images of the wound scratch at different time intervals should be presented.

- Where is the loading control to normalize protein levels for the immunoblotting assay? Please include the raw image files as supplementary files.

- Why have the authors not considered studying the combination effect of more than one ECM?

- I would suggest considering another title for the manuscript to avoid similarity with your previously published abstract "The role of extracellular matrix proteins on placental angiogenesis in severe fetal growth restriction."

- Textually, I think the manuscript would benefit from removing some rather long sentences.

- Further details about the mechanism of ECM's effect on angiogenic characteristics should be included in the discussion section.  

In conclusion, this is an exciting paper suitable for the journal but after a good revision.

Author Response

Thank you for your review. Your comments have helped us substantially improve this manuscript. Responses to your individual comments are below.

The migrational and angiogenic properties of ECs should be investigated extensively to better elucidate how extracellular matrix proteins regulate angiogenesis beyond just the proliferation activity. For example, relative angiogenic gene expression and morphological characterization of vasculogenic networks should all be considered to strengthen this work.

  • We agree that assessment beyond proliferation and migration are important. However, in this particular case, given the lack of substantial effects of ECM proteins, we opted not to further pursue investigation of angiogenic gene expression.
  • We also did not pursue morphological characterization of vasculogenic networks, which we have interpreted to mean tube formation assays, as we believe that “spiking” a matrix generated by Engelbroth-Holm-Swarm sarcoma cells (i.e. Matrigel®) that is typically already rich in laminin, collagens, and other ECM proteins would not allow for elucidation of the effects of individual ECM proteins.
  • However, we acknowledge your very important point, and in the future, we plan to continue better characterizing the complexities of placenta stroma in both a 2D and 3D manner. At this point, we will definitely be adding in evaluation of angiogenic gene expression as well as other biologically relevant morphologic characterization such as luminal formation.

The same group previously reported that integrin dysregulation contributes to the reduced angiogenic potential of FGR cells. Why were integrins not evaluated in the present study?

  • We acknowledge that integrin-ECM interactions are critical. However, the goal of this study was to characterize the specific effects of individual ECM proteins on the angiogenic properties of control and severe FGR ECs. In the absence of identifying a specific substrate that differentially regulates EC angiogenic properties, we did not feel that delving into all the various integrin heterodimers that bind to the different ECM proteins was within the scope of this manuscript. However, as our laboratory continues to learn more about the composition, architecture, stiffness, and remodeling of placental villous stroma, we anticipate that we will better hone in on key integrin heterodimers (and other surface membrane proteins such as receptor tyrosine kinases) that regulate control and FGR EC angiogenic properties.

For the wound scratch assay, representative images of the wound scratch at different time intervals should be presented.

  • Representative wound scratch images have been added for ECM substrates that demonstrated significant differences in supplemental data.
  • Migratory capacity was calculated via relative wound density, defined as the proportion of the initial wound scratch occupied by cells, while also accounting for changes in background, non-wound, and wound density, at each time point. This method controls for potential differences in wound scratch size and any variation in confluence at the time the scratch is generated.

Where is the loading control to normalize protein levels for the immunoblotting assay? Please include the raw images as supplementary files.

  • We are sorry that we did not clearly describe apoptosis findings in our initial manuscript. Our goal had been to depict that there was absence of cleaved caspase-3 in all of the conditions (except the positive control), demonstrating the absence of apoptosis under all substrate conditions. In contrast, total caspase-3 was present, as anticipated, in all specimens. As total caspase-3 was expressed in all lanes, we did not initially show the total protein normalization that is part of the JESS SimpleWestern capillary immunoblot system. However, we have added this to both the cleaved and total caspase-3 images to show the raw images.

Why have the authors not considered studying the combination effect of more than one ECM?

  • Thank you for this very important comment. We agree that villous stromal matrix composition is certainly more complex than any individual ECM protein. Because we had previously found that matrix generated from severe FGR placental fibroblasts displayed decreased, disordered ECM deposition of collagen I and fibronectin, as compared to matrix from control placental fibroblasts, the goal of this manuscript was to evaluate whether the presence of individual ECM substrates including collagen I and fibronectin were sufficient to alter severe FGR EC angiogenic properties.
  • Our laboratory is currently studying composition, architecture, and stiffness of villous stromal ECM in more detail. Findings from these experiments will inform future experiments that investigate villous stromal matrix in its more biologically relevant in vivo form.

I would suggest considering another title for the manuscript to avoid similarity with your previously published abstract “The role of extracellular matrix proteins on placental angiogenesis in severe fetal growth restriction.”

  • Based on your suggestion, we have changed the title to “Angiogenic function of human placental endothelial cells in severe fetal growth restriction is not rescued by individual extracellular matrix proteins.”

Textually, I think the manuscript would benefit from removing some rather long sentences.

  • We apologize for this and acknowledge the tendency to use long sentences. We have attempted to modify the manuscript for ease of readability.

Further details about the mechanisms of ECM’s effect on angiogenic characteristics should be included in the discussion section.

  • This is a wonderful suggestion, and the discussion has been modified.

Reviewer 5 Report

Dear authors this is a very important study evaluating the potential role of several ECM adhesion proteins on EC from fetal tissues of both normal and FGR samples. However some concerns should addressed by you in order to improve this manuscript.

1. Please add a figure indicating the design of the study.

2. The statistical methods used for data analysis of EC proliferation and migration should be evaluated by using a GLMM or a GLM in order to evaluate the effect of the factors studied and their potential interactions indicating clearly in which time there were significant differences. 

Author Response

Thank you for your review. Your comments have helped us substantially improve this manuscript. Responses to your individual comments are below.

Please add a figure indicating the design of the study.

  • This has been added as Figure 1.

The statistical methods used for data analysis of EC proliferation and migration should be evaluated by using a GLMM or GLM in order to evaluate the effect of the factors studied and their potential interactions indicating clearly in which time there were significant differences.

  • Thank you for this important comment. At the initial time of analyses, we debated amongst our team as to the ideal statistical test. We initially chose to pursue non-linear regression to compare the curves as we believed that overall rates of proliferation and wound closure were more biologically relevant than the potential for significant differences at specific time points.
  • However, and based on your critique along with those of other reviewers, we have re-analyzed proliferation and migration data using two-way ANOVA with repeated measures. This has led to some specific differences in the results as compared to our original submission. The methods, results, associated figures, and discussion have been revised, accordingly.

Round 2

Reviewer 1 Report

Most concerns raise in 1st round of review have been addressed.

Minor Concerns:

1.     Statistical analysis: As mentioned by the response from the authors the sample size of this study is small. Hence, non-parametric method instead of ANOVA need to be used for statistical analysis.

Author Response

Thank you for reviewing our resubmission. We greatly appreciate your feedback and acknowledge the additional minor concern that was brought up in your review of the resubmission:

  • To determine sample size, we performed sample size calculations for each outcome with continuous variables, and we chose the largest sample size that could be utilized for all the studies. Specifically, we found that 8 subjects per group were required for 90% power (a=0.05) to detect a 10% difference in average migratory capacity. This difference was chosen because we felt that it was biologically significant and because it was the most conservative effect size, thereby increasing the sample size estimation.
  • We apologize for the inadvertent omission in the manuscript, but data with continuous variables were also analyzed for distribution. While we acknowledge that normality tests have less power to detect non-Gaussian distributions in small populations, Shapiro-Wilk testing did not suggest a non-normal distribution for either control or FGR subjects. Because of this and the fact that parametric tests have greater power to detect if an effect truly exists, we opted to use parametric testing for our continuous data. This has been updated in the manuscript.

Reviewer 5 Report

Dear authors, all my concerns were addressed.

Author Response

Thank you for reviewing our resubmission. Your feedback after our first submission helped us substantially improve the manuscript.